# Transcriptomic and Phenotypical Analysis of the Physiological Plasticity of *Chamaecyparis hodginsii* Roots under Different Nutrient Environments and Adjacent Plant Competition

**DOI:** 10.3390/plants13182641

**Published:** 2024-09-21

**Authors:** Bingjun Li, Wenchen Chen, Yanmei Pan, Wenxiu Wu, Ying Zhang, Jundong Rong, Tianyou He, Liguang Chen, Yushan Zheng

**Affiliations:** 1College of Forestry, Fujian Agriculture and Forestry University, Fuzhou 350002, China; fafulbj@163.com (B.L.); cerdwin2003@163.com (W.C.); panyanmei@163.com (Y.P.); wuwenxiu0929@126.com (W.W.); 17350229322@163.com (Y.Z.); rongjd@126.com (J.R.); clguang_cn@163.com (L.C.); 2College of Landscape Architecture and Art, Fujian Agriculture and Forestry University, Fuzhou 350002, China; hetianyou1985@163.com

**Keywords:** *Chamaecyparis hodginsii*, heterogeneous nutrient environment, plant competition, physiological plasticity, transcriptome

## Abstract

*Chamaecyparis hodginsii* seedlings undergo significant changes during growth due to different nutrient environments and adjacent plant competition, which is evident in the physiological plasticity changes in their roots. Therefore, in this experiment, 20 one-year-old elite *C. hodginsii* family seedlings were selected as the test objects, and the different nutrient environments and adjacent plant competition environments in nature were artificially simulated. Four nutrient environments (N heterogeneous nutrient environment, P heterogeneous nutrient environment, K heterogeneous nutrient environment, and homogeneous environment) and three planting patterns (single plant, conspecific neighbor, and heterospecific neighbor) were set up to determine the differences in root physiological indexes and plasticity of different family seedlings, and the families and treatment combinations with higher comprehensive evaluation were selected. The transcriptome sequencing of fine roots of *C. hodginsii* under different treatments was performed to analyze the differentially expressed genes. The results showed that the root activity, antioxidant enzyme activity, and nutrient element content of *C. hodginsii* seedlings in the N and P heterogeneous environments were higher than those in the homogeneous nutrient environment, while there was no significant difference between the K heterogeneous nutrient environment and the homogeneous environment, but MDA content was higher than that in other nutrient environments. The root activity and antioxidant enzyme activity in the competitive patterns were generally higher than those in the single plant and reached the peak in the heterospecific neighbor. The root physiological plasticity index of line 490 was the highest, but the comprehensive evaluation of root physiological indexes of lines 539 and 535 was better. The pattern with the highest comprehensive evaluation score was P heterogeneous nutrient environment × heterospecific neighbor. The effects of the N and P heterogeneous nutrient environments on root transcriptome genes were similar, which significantly increased DNA transcription and regulatory factor activity, while K heterogeneous nutrient environment focused on the regulation of root enzyme activity. The heterogeneous nutrient environment induces the conduction of hormone signals in the roots of *C. hodginsii* and induces the synthesis of phenylpropanone. The biosynthesis of phenylpropanone in the roots of *C. hodginsii* will increase significantly under competitive patterns. In summary, the N and P heterogeneous nutrient environments and the heterospecific neighbor can improve the root physiological indexes of *C. hodginsii* families, and the root physiological indexes of lines 539 and 535 are the best. The nutrient environment and competition pattern mainly affect the root system to transmit hormone signals to regulate enzyme activity.

## 1. Introduction

Plants must continuously maintain root growth to explore nutrient-rich areas in the soil and obtain nutrients to the maximum extent in a heterogeneous nutrient environment. A series of morphological and physiological responses of roots ensures plants obtain nutrients within patches [1]. Furthermore, the differences in the physiological plasticity of plant roots often determine their adaptability to different heterogeneous nutrient environments. Roots of different *Cunninghamia lanceolata* lines proliferate extensively in nutrient-poor patches, and the phosphorus content in the roots of nutrient-poor patches is higher than that in nutrient-rich patches. Heterogeneous nutrient environments can promote root development and adjust element allocation ratios [2,3]. Significant tolerance differences in plant roots to N, P, and K were also observed. Moreover, environments with excessive N can hinder the growth of some plants and even cause plant death [4]. 

Additionally, plant adaptability to heterogeneous nutrient environments is related to genes. Zhao et al. showed that in poplar, overexpressing *OsPTR9* significantly improved the tolerance of poplar plants to low nitrogen stress and increased the mass factions of the N, P, and K stored in poplar leaves [5]. Wang et al. showed that *ZmMYB12* is involved in the responses to low phosphorus stress and improves the absorption and utilization efficiency of phosphorus elements in plants [6]. In recent years, plant response to spatial heterogeneity of the available soil resources and the adaptive measures of plant roots to heterogeneous nutrients have become hotspots in forest cultivation research. Most current studies have only compared root morphology responses [7,8]. However, less attention is paid to the differences in plant root vitality, nutrient content, absorption efficiency in different heterogeneous nutrient environments, and transcriptomic profiles of plant roots in heterogeneous nutrient environments.

Competition is an important phenomenon during plant growth [9,10], and adjacent plant roots inevitably influence plants when exploring heterogeneous nutrients [11]. Previous reports have shown that rapidly growing plants can absorb or deplete most of their nutrients and light resources in a nutrient patch, thereby inhibiting the growth of adjacent plants. Tree species with high competitiveness can regulate their physiological activity and the nutrient storage morphology in the organs, separating the ecological niches. The result is better utilization of nutrient resources in different ecological regions, improved utilization rates of environmental resources, and co-existence between species [12]. 

Changes in the plasticity of plants in adjacent plant competition play an important role in the efficient utilization of nutrient resources and productivity improvement. Plants with high morphological and physiological plasticity can absorb nutrients and water from the soil more quickly, thus promoting plant growth and gaining a competitive advantage [13]. Most forest soil nutrients exhibit heterogeneity; thus, plant root responses to adjacent plant competition intensity vary significantly in different heterogeneous nutrient environments. Moreover, the competition relationship between adjacent forest plants is also more complex [14]. In summary, when plants face heterogeneous nutrient environments and competition from adjacent plants, their roots exhibit corresponding changes according to the environment and competition type, enabling them to better seek nutrients and water. The trend of changes in plant roots during this process and the regulatory mechanisms of these two environmental factors have been great research interests in recent years [15].

*Chamaecyparis hodginsii* is a tree species with tolerance to barren soil, shallow rooting, developed lateral roots, and no obvious main roots. *C. hodginsii* can be used as a pioneer tree for cultivating barren land and a mixed-planting tree for forest plantations. However, there is a lack of research on *C. hodginsii* in this research field. Different tree species exhibit significant differences when facing heterogeneous nutrient environments and competition from adjacent plants [16,17]. Therefore, this study used 20 *C. hodginsii* family seedlings as experimental materials under different heterogeneous nutrient environments and types of adjacent plant competition to investigate the root vitality and element content of *C. hodginsii* seedlings. At the same time, transcriptome sequencing (RNA-seq) of the fine roots of *C. hodginsii* identified the differentially expressed genes (DEGs). GO (Gene Ontology) and KEGG (Kyoto Encyclopedia of Genes and Genomes) enrichment analysis annotated the functions of the DEGs and determined the key regulatory genes responsible for adapting *C. hodginsii* to heterogeneous nutrient environments and adjacent plant competition. 

## 2. Materials and Methods

### 2.1. Overview of the Experimental Site

The experimental site was in the greenhouse (119°14′47.37″ E, 26°05′29.88″ N) of the College of Landscape Architecture and Art, Fujian A&F University (left panel of Figure 1). This greenhouse is an experimental (educational) facility with good ventilation, spray cooling equipment, and sunscreen (right panel of Figure 1). During the experiment, the greenhouse temperature was 14–30 °C (average 25 °C); the relative humidity during the growing season was between 78 and 83% (average 80%); and the daily sunshine duration was 6:00–18:00 (average 12 h).

### 2.2. Experimental Materials

Twenty *C. hodginsii* family seedlings (the age of family seedlings is one year old) with superior growth characteristics were selected as the experimental materials. Table 1 shows the seedling line numbers (germplasm original accession numbers). These lines were cultivated in the early growth stage of the Fengtian state-owned forest farm. In early March 2022, the seedlings were transplanted to the College of Landscape Architecture and Art greenhouse at Fujian A&F University. The average basal diameter of the selected *C. hodginsii* seedlings was 2.65 ± 0.86 mm, and the average height of the seedlings was 21.47 ± 4.12 cm to ensure consistent initial conditions for the experimental materials. Thus, 90 *C. hodginsii* seedlings were selected for each line, and pot experiments were set up with nitrogen (N), phosphorus (P), and potassium (K) heterogeneous and homogeneous nutrient environments. The pot culture medium was collected from the barren acidic red soil on the back hill of Fujian A&F University, with a 5.91 g.kg^−1^ organic matter content and 0.43 and 0.40 g.kg^−1^ total nitrogen and total phosphorus contents, respectively. The contents of hydrolyzed nitrogen, available K, and available P were 29.08, 238.68, and 6.15 mg.kg^−1^, respectively, with a pH of 4.97.

### 2.3. Experimental Design

The experiment involved a 4 × 3 double-factor factorial design (four nutrient environments and three planting patterns) planted in polyethylene pots. The experiment started in early March 2022. Barren, acidic red soil was loosened and then disinfected with 0.5% potassium permanganate (this measure can effectively prevent plant wilt, stem blight, sudden wilt, and root rot). The soil was then covered with plastic film, sealed, exposed to air, and dried for a week before screening. The upper end of the container served as a buffer layer. The soil was filled to 4 cm below the container top. In contrast, the lower part of the container was divided into three parts: the nutrient-rich patch (A side), the nutrient-poor patch (B side), and the nursery planting area located in the middle part (this part was the initial growth area of the seedlings. The soil is the substrate soil for normal potted plants (i.e., without any fertilizer applied) (Figure 2). The two nutrient patch areas had the same volume, and the nutrient patches were separated from the nursery planting area by a non-woven fabric coated with agar on the surface. This separation material prevented nutrient leakage and loss, which could affect the results. This non-woven fabric also ensured a smooth root penetration, facilitating the observation of root growth in the later stage. Each pot contained approximately 4.5 kg of soil. 

After a large number of preliminary experiments and reference to the results of previous studies [2,17], it is concluded that the concentrations of N, P, and K in the fertilization and seedling stages of *C. hodginsii* did not exceed 110, 300, and 190 mg.kg^−1^, respectively. The nutrient element ratio of these three important nutrient elements was N:P:K = 2:5:3, and the N, P, and K concentrations in the nutrient patches on both sides were approximately 50, 125, and 75 mg.kg^−1^ [18] (Table 2). Furthermore, urea (N 46%), calcium superphosphate (P_2_O_5_ 16%), and potassium chloride (K_2_O 60%) were added to each kg of medium on both sides of the container. One side of the container was filled with nutrient-rich soil with corresponding nutrient elements and fertilized with twice the amount of that element so that the corresponding content of the elements was twice that of the homogeneous patches. However, nutrient-poor patches did not receive the corresponding nutrient elements. The detailed recipe for nutrient application is shown in Table 1. In treating different heterogeneous nutrient environments, the A side of the pot represented the relatively abundant nutrient elements in the heterogeneous environment, and the B side represented the relatively poor nutrient elements. The nutrient contents of the A and B sides of the homogeneous nutrient environment were the same.

Additionally, three planting patterns were adopted in different nutrient environments, including (1) the single-plant pattern, (2) the *C. hodginsii–C. hodginsii* planting pattern providing a conspecific neighbor, and (3) the *C. hodginsii–Cunninghamia lanceolata* planting pattern providing a heterospecific neighbor. (*Cunninghamia lanceolata* as the most common and suitable hybrid species for *C. hodginsii* has been confirmed by most research institutes, so we chose to use a mixture of *Cunninghamia lanceolata* and *F. hodginsii* seedlings to simulate interspecific competition). One *F. hodginsii* seedling was planted in the middle of each pot in the single-plant pattern. In the conspecific neighbor and heterospecific neighbor patterns, two seedlings were planted at two opposite points on the two sides of the midline. 

They were cultivated in N, P, and K nutrient heterogeneous and homogeneous environments with six pots for each treatment, totaling 1440 pots. On average, about 100 mL of water was poured every day. In order to ensure the consistency of nutrient patch environment proofreading, the changes in nutrient content on both sides of all pots were measured regularly in the middle of each month. In order to maintain the heterogeneity of soil nutrients, fertilization was carried out again in early September 2022 and early January 2023. The amount of fertilization was calculated according to the current nutrient content of each patch. The fertilizer was the same as the first fertilization, so the overall heterogeneity remained unchanged.

### 2.4. Indicators and Measurements

#### 2.4.1. Determination of Root Vitality, Antioxidant Enzyme Activity, MDA Content, and Nutrient Elements

Plants for the experiment were harvested in April 2023, and the height and basal diameter of each treated plant were measured. The pots with selected *C. hodginsii* seedlings were cut open from the outside with scissors. The entire *C. hodginsii* seedling was removed, and the soil attached to the roots was washed off with tap water. Secondary cleaning was done using distilled water, and the water on the root surface was wiped dry using absorbent paper. Finally, the non-woven fabric was cut open to preserve the complete root tissue as much as possible. The roots were cut off and sealed in a self-sealing bag, placed in an ice box, and brought back to the laboratory for immediate treatment at a low temperature to minimize the root vitality, and antioxidant enzyme activity was not affected.

All treated *C. hodginsii* root tissues were placed in an insulated ice-filled box for the following experiments. A total of 15 root tips or white roots of *C. hodginsii* under each treatment were collected for root vitality and antioxidant enzyme activity measurements. Root vitality was measured using the TTC staining method (2,3,5-triphenyltetrazolium chloride) [19]. Superoxide dismutase (SOD), peroxidase (POD), and catalase (CAT) activities were measured using the SOD, POD, and CAT assay kits (Suzhou Comin Biotechnology), respectively. 

The thiobarbituric acid (TBA) colorimetric method determined the content of malondialdehyde (MDA) [20]. *C. hodginsii* root systems of seedlings in each pot were cut off with scissors, washed, dried, placed into kraft paper, and subsequently placed in an oven at 105 °C for 15 min for desiccation. The desiccated samples were dried at 85 °C to a constant weight. Roots were ground to powder using an ultra-high-speed crusher, passed through a 0.15 mm sieve, and stored in a dryer for further experiments. Six pots for each line were treated with total N and total C, and carbon–nitrogen ratios (C/N) were measured using an Element Micro Analyzer (VELP, State for European, Germany). The total P content of the *C. hodginsii* root system was determined using the concentrated H_2_SO_4_-H_2_O_2_ digestion molybdenum antimony colorimetric method [21].

#### 2.4.2. RNA Extraction and Transcriptome Sequencing

One-year-old *C. hodginsii* potted seedlings were used as experimental materials. The roots of line 495 plants with the closest growth to the average value were used for the heterogeneous nutrient environment and adjacent plant competition, ensuring uniformity of the experimental materials. This treatment method was adopted from the experimental design in Section 2, and the treatment time was one year. Three plants with normal growth for each treatment were selected and harvested in April 2023 upon the end of the experiment. Each potted container was cut in half with a blade, the entire *C. hodginsii* seedling was removed, and the soil attached to the roots was washed off with tap water. The roots were cleaned again using distilled water, and the excess water on the root surface was absorbed with absorbent paper. Eight to fifteen fine roots (white roots) with a low degree of lignification were collected from each *C. hodginsii* seedling, cut into segments with a blade, placed in a 10 mL centrifuge tube, and immersed immediately in liquid nitrogen. All samples were stored in a −80 °C refrigerator until further use.

RNA was extracted from three biological replicates of each treatment using the Novogene kit (Beijing, China). Next, RNA integrity was determined using the Agilent 5400 bioanalyzer (Agilent Technologies, Santa Clara, CA, USA) [22]. cDNA libraries were constructed and sequenced using the NovaSeq 6000 Illumina high-throughput sequencing platform (Illumina, San Diego, CA, USA) (Table 3).

### 2.5. Data Analysis

The physiological plasticity index derived from physiological indicators of the root system reflects the adaptability of plant roots to different environments. The formula is *I*_SI_ = (V_max_ − V_min_)/V_max_, where *I*_SI_ represents the physiological plasticity index; V_max_ and V_min_ are the maximum and minimum values of a specific indicator, respectively. The physiological plasticity index of the *C. hodginsii* root system is the ratio of the difference between the maximum and minimum values of each indicator and the maximum value of *C. hodginsii* seedlings in each family. 

Statistical analysis of the data was conducted using SPSS 22.0 software. One-way ANOVA and a graph-based approach were used to determine the differences in various indicators under different treatments (α = 0.05). Three-factor analysis of variance was used to analyze whether there was an interaction between planting patterns, nutrient heterogeneity, and family differences in the growth and leaf phenotypic traits of *C. hodginsii*. Principal component analysis (PCA) was used to obtain the dominant factors, and the combination of different environmental factors was comprehensively sorted. The membership function method was used to comprehensively evaluate and screen the families with better growth and leaf phenotypic traits, and the Origin2024 software was used for mapping.

## 3. Results

### 3.1. The Effect of Different Treatments on the Root Activity of Various Families of C. hodginsii

The effects of nutrient environment and adjacent plant competition on the root activity of different *C. hodginsii* families were significantly different (Figure 3). The root activity in the N and P heterogeneous nutrient environments was higher than that in the K heterogeneous and homogeneous nutrient environments, and the root activity of lines 535 and 455 was higher in the N and P heterogeneous nutrients. The root activity in the K heterogeneous environment was generally lower than that in the homogeneous nutrient environment. The root activity of *C. hodginsii* families in the competition patterns was higher than that in the single-planting pattern, and the root activity of *C. hodginsii* families in the heterospecific neighbor was higher than that in the conspecific neighbor. Among them, the root activity of lines 535 and 539 was generally higher in the competition patterns. There were significant differences in the root activity of *C. hodginsii* families in different planting patterns, indicating that the competition patterns could effectively improve the root activity of *C. hodginsii* families.

### 3.2. Effects of Different Treatments on Antioxidant Enzyme Activity and MDA Content in the Roots of Various C. hodginsii Lines

Heterogeneous nutrient environments and adjacent plant competition significantly affected the antioxidant enzyme activity and MDA content in the root systems of different *C. hodginsii* lines (Figure 4). The average root CAT activity of most *C. hodginsii* seedlings in the P heterogeneous nutrient environment reached its peak. Line 535 had the highest CAT activity, which was 0.3%, 38.6%, and 31.1% higher than that in the N and K heterogeneous and homogeneous nutrient environments, respectively. In the N heterogeneous nutrient environment, the average root CAT activity of each line was higher than that in the homogeneous environment but slightly lower than that in the P heterogeneous nutrient environment. In the K heterogeneous nutrient environment, the average CAT activity of most lines was slightly higher than that in the homogeneous environment, but the difference was insignificant. From the perspective of the impact of planting patterns on *C. hodginsii* family seedlings, the average CAT activity of *C. hodginsii* seedlings in each line peaked in the heterospecific neighbor, with line 535 having the highest CAT activity, which was 17.4 and 0.4% higher than that of the single and the conspecific neighbor, respectively. The average CAT activity of all lines in the conspecific neighbor was also higher than that of the single-plant pattern, indicating that the competition pattern promotes CAT activity in the roots of *C. hodginsii*.

The root POD activities of various *C. hodginsii* families were similar to the CAT activity trend in heterogeneous nutrient environments. The average POD activity peaked in the P heterogeneous nutrient environments, with line 539 having the highest POD activity at 2012.22 U.g^−1^, which was 50.6%, 74.3%, and 18.6% higher than in the N, K heterogeneous, and homogeneous environments. The root POD activity in the N heterogeneous nutrient environments was higher than that in homogeneous nutrient environments. However, the average POD activity of each line was consistent in both K heterogeneous and homogeneous nutrient environments, but the overall difference was insignificant. The average POD activity of each line in the three planting patterns peaked in the heterospecific neighbor, among which the average value of POD activity in lines 539, 464, and 500 was higher, and the average value of POD activity in the conspecific neighbor was higher than that in the single plant.

The SOD activity in the roots of *C. hodginsii* seedlings was generally higher in the N and P heterogeneous environments than in the homogeneous environment. The average value of SOD activity in lines 535 and 539 was higher, and the average value of SOD activity in the N and P heterogeneous environments was 271.84 U.g^−1^ and 277.11 U.g^−1^, respectively. The average value of SOD activity in the K heterogeneous environment was also higher than in the homogeneous nutrient environment, indicating that the heterogeneous nutrient environment could effectively improve the SOD activity of *C. hodginsii* roots. The average SOD activity in the root systems of all the lines in the heterospecific neighbor was higher than that in the single plant and the conspecific neighbor. Among them, line 535 had 355.01 U.g^−1^, which was 72.5% and 59.0% higher than in the single plant and the conspecific neighbor. Most lines in the conspecific neighbor had slightly higher SOD activity than those in the single-plant pattern, but the difference was insignificant. This result indicates that competition promoted SOD activity in the root systems of *C. hodginsii* seedlings.

The MDA content of roots in the P heterogeneous nutrient environment and the homogeneous nutrient environment did not show a significant change, while the MDA content of roots in the K heterogeneous nutrient environment was higher than that in other nutrient environments, indicating that the K heterogeneous nutrient environment would significantly increase the MDA content in the roots of *C. hodginsii*. The average MDA content of roots in the conspecific neighbor was higher than that in the single plant and the heterospecific neighbor, and the average MDA content of line 485 was the highest, which was 37.0% and 10.5% higher than that in the single plant and the heterospecific neighbor, respectively. The average MDA content in the single plant was slightly lower than that in the heterospecific neighbor, but the difference was not significant.

### 3.3. The Effect of Different Treatments on the Nutrient Element Contents in the Roots of Various C. hodginsii Lines

Figure 5 suggests significant differences in the effects of heterogeneous nutrient environments and adjacent plant competition on the nutrient contents of root systems in different *C. hodginsii* families. The average C content in the roots of *C. hodginsii* families peaked in the P heterogeneous nutrient environment. Line 539 had the highest C content (118.681 mg.g^−1^), which was 37.4%, 45.7%, and 52.1% higher than in the N, K heterogeneous, and homogeneous nutrient environments, respectively. The average C content in the roots of N heterogeneous nutrient environments was generally higher than in the K heterogeneous nutrient environments. However, the overall averages were higher than those in homogeneous nutrient environments. Thus, heterogeneous nutrient environments promoted C accumulation in the roots of most *C. hodginsii* lines. Considering the impact of planting patterns on the seedlings of *C. hodginsii* families, the average root C content of most lines peaked in the heterospecific neighbor. Line 490 reached 108.046 mg.g^−1^, which was 26.5% and 31.0% higher than in the single plant and conspecific neighbor, respectively. However, the overall difference in the C content among most families in different planting patterns was insignificant.

The average root N content peaked in the P heterogeneous nutrient environment. Line 506 reached a maximum of 3.051 mg.kg^−1^, which was 25.4%, 50.4%, and 65.1% higher than in the N heterogeneous, K heterogeneous, and homogeneous nutrient environments, respectively. The root N content of each line showed a relatively non-uniform trend in the N and K heterogeneous nutrient environments, but the overall difference was insignificant. The average values of the two heterogeneous nutrient environments were also higher than in the homogeneous nutrient environment, indicating that the heterogeneous nutrient environment promoted N accumulation in the roots of most *C. hodginsii* families. The impact of the planting pattern on the seedlings of *C. hodginsii* families increased the root N content in the competitive patterns compared to the single-plant pattern. The overall difference between the heterospecific neighbor and the conspecific neighbor was insignificant. Line 506 recorded the highest impact (2.421 mg.kg^−1^) in the conspecific neighbor, and the highest value in the heterospecific neighbor was 2.618 mg.g^−1^ in line 454.

The P content of roots in different nutrient environments showed a similar trend to that of C and N, and all reached the peak in the P heterogeneous nutrient environment. The highest P content of line 454 was 0.720 mg.kg^−1^, which was 29.0%, 55.2%, and 95.5% higher than in the N, K heterogeneous, and homogeneous nutrient environments, respectively. The P content of roots in the N heterogeneous environment was higher than that in the K heterogeneous and homogeneous nutrient environments. In different planting patterns, the content of P element in the roots of each family of *C. hodginsii* showed a similar trend with the content of N element, and the content of P element in the roots under the competition patterns was generally higher than that under the single plant.

Under different nutrient environments and planting patterns, the root C/N of *C. hodginsii* families did not show a relatively uniform trend (Figure 6). Among them, the highest was line 501 in the homogeneous nutrient environment at 87.704, and the lowest was line 535 in the N heterogeneous nutrient environment at 30.852. The root C/N of each family in different planting patterns also did not show a relatively uniform change trend; among these, the highest was line 490 in the single plant at 61.729, and the lowest was line 490 in the conspecific neighbor at 34.937, but the overall C/N average value was the highest in the single plant.

The root N/P ratios of various *C. hodginsii* lines showed a relatively non-uniform trend under different heterogeneous nutrient environments and planting patterns. However, the overall average root N/P ratios in the K heterogeneous nutrient environments reached 5.038, 10.6%, 20.6%, and 3.1% higher than those in the N and P heterogeneous and homogeneous nutrient environments. Moreover, the root N/P of most *C. hodginsii* families peaked in the single-plant planting pattern, with line 540 reaching a maximum (6.634), which was 32.0% and 41.9% higher than those of the conspecific neighbor and the heterospecific neighbor. Therefore, the root N/P of most *C. hodginsii* families in the non-competitive pattern was higher than that in competitive patterns.

### 3.4. Analysis of Interaction between Different Treatments and Families

The results of the three-factor analysis of variance showed that except for MDA content and C/N, planting patterns had significant or extremely significant effects on most root physiological indexes (Table 4). Except for N/P, the nutrient environment had significant or extremely significant effects on other root physiological indexes. The differences between *C. hodginsii* families had significant or extremely significant effects on all physiological indexes of roots. There were significant interactions between planting patterns and nutrient environment on root activity, MDA content, C content, and P content of *C. hodginsii*. There were significant or extremely significant interactions between planting patterns and family differences for root activity, CAT activity, POD activity, MDA content, C content, and P content. There were significant or extremely significant interactions between the nutrient environment and family differences for root activity, CAT activity, POD activity, SOD activity, C content, and the N content of *C. hodginsii*. There were significant or extremely significant interactions among planting pattern, nutrient environment, and family differences for root activity, CAT activity, POD activity, and the MDA content of *C. hodginsii*.

### 3.5. Physiological Plasticity Indexes of the Root System of C. hodginsii Families 

Lines 490, 535, and 539 had a higher root physiological plasticity index of *C. hodginsii* in each family, and the plasticity index was 0.637, 0.611, and 0.643, respectively. The lowest was 0.508 in line 506 (Table 5). It can be seen that the overall difference in root physiological plasticity index between families is small. The results showed that the average plasticity indexes of P content, root activity, and the MDA content of *C. hodginsii* seedlings were higher, at 0.688, 0.683, and 0.653, respectively. The average plasticity indexes of C content and CAT activity were relatively low, at only 0.463 and 0.412, respectively.

### 3.6. Comprehensive Evaluation of C. hodginsii Seedlings

The ten physiological indicators of the roots of 20 *C. hodginsii* lines were normalized using membership functions, generating five weight coefficients to calculate the comprehensive score values of each family. It can be seen from Table 6 that the comprehensive evaluation of root physiological indexes of line 539 was up to 0.806, followed by line 535, and the comprehensive evaluation of the root physiological indexes of lines 467 and 536 was low, at only 0.304 and 0.339, respectively. Combined with the results of the cluster analysis of different families (Figure 7) and using the sum of squares of deviations method, the Euclidean distance was used as the genetic distance. When the genetic distance was 0.05, the 20 families of *C. hodginsii* could be divided into four categories. Among them, lines 535 and 539 represented group I, and the main characteristics were that the root physiological indexes, such as root activity, antioxidant enzyme activity, and nutrient content, were higher. The II group was mainly represented by 14 families, such as lines 455, 474, and 540, accounting for the largest part of the overall family, and its indicators were generally closer to the average level; group III was mainly represented by line 536, its comprehensive evaluation of various indicators was low, and the root physiological indicators were generally poor. Group IV is mainly represented by three families of lines 454, 464, and 467, and its root physiological indexes are generally lower than the average level. Based on the comprehensive evaluation results of the root physiological indexes of different families of *C. hodginsii*, lines 535 and 539 were finally selected as the seedlings of *C. hodginsii* families with better physiological performance of roots under different treatments.

### 3.7. Principal Component Analysis of the Root System Vitality of C. hodginsii under Different Treatments

Figure 8 and Table 7 show the principal component analysis of the ten physiological indicators of *C. hodginsii* root systems, identifying two main components. The cumulative variance contribution rate of the first principal component was 69.521%, and that of the second principal component was 14.284%. The cumulative variance contribution rate of the first two principal components was 83.805%, suggesting they could explain all the data variations. The absolute coefficients of root P content, CAT activity, and SOD activity in the first principal component were relatively large. Therefore, they were the most important indicators reflecting the physiological activity of the *C. hodginsii* root system in different planting patterns and heterogeneous nutrient patches.

The eigenvalues and eigenvectors are the two principal component expressions, as follows:Y_1_ = 0.352X_1_ + 0.363X_2_ + 0.349X_3_ + 0.357X_4_ − 0.098X_5_ + 0.293X_6_ + 0.322X_7_ + 0.372X_8_ − 0.179X_9_ − 0.355X_10._

Y_2_ = − 0.266X_1_ − 0.051X_2_ − 0.193X_3_ − 0.059X_4_ + 0.633X_5_ + 0.224X_6_ + 0.365X_7_ + 0.085X_8_ − 0.527X_9_ + 0.131X_10._


Table 8 shows the standardized raw data substituted into the principal component expression to calculate the comprehensive principal component scores for each treatment. The patterns with higher comprehensive evaluation scores were P heterogeneous nutrient environment × mixed planting pattern and N heterogeneous nutrient environment × mixed planting pattern. The root physiological indicators of *C. hodginsii* seedlings in the N and P heterogeneous nutrient environments were better than those in the K heterogeneous and homogeneous nutrient environments. In contrast, the root physiological indicators were more conducive to improving the growth vitality and nutrient element accumulation ability of *C. Hodginsii*. The mixed planting pattern was more conducive to *C. hodginsii* root growth than the unmixed and single-plant planting patterns. Additionally, the competitive patterns generally improved the physiological activity intensity of the root system.

### 3.8. Transcriptome Sequencing of the C. hodginsii Root System under Different Treatments

The overall concentration of RNA samples was around 13 ng.μL^−1^–71 ng.μL^−1^, and the average integrity was 8.3, indicating high RNA quality, which satisfies the quality requirements for subsequent library construction and sequencing (Table 9). 

There were low-quality reads [23] (with base numbers of Qphred ≤ 5 accounting for more than 50% of the entire length), adapters, and high content of unknown bases (Table 9). The number of clean reads in the samples ranged from 43.69 Mb to 54.12 Mb, accounting for over 95% of the total reads. The Q20 value was approximately 97.32, the Q30 value was >91%, and the overall error rate was approximately 0.01, indicating a low error rate and high-quality sequence data. The clean reads were mapped to the reference genome (Table 10) with a 78.54–92.36% mapping rate. The mapping rates of the homogeneous and P heterogeneous nutrient environments were high (>87%), indicating a complete reference genome assembly with no contamination from the relevant experiments. Therefore, the sequencing quality was high, and the sequencing data could be further analyzed.

The (|log2 (FoldChange)| ≥ 1 & padj ≤ 0.05) differential gene identification criteria revealed 2187, 2157, 3805, 1577, 3787, and 1793 significant DEGs between the N heterogeneous and homogeneous nutrient environments (A-N vs. A-T), P heterogeneous and homogeneous nutrient environments (A-P vs. A-T), K heterogeneous and homogeneous nutrient environments (A-K vs. A-T), pure and single-plant planting patterns (B-C vs. B-D), mixed and single-plant planting patterns (B-H vs. B-D), and the unmixed and mixed planting patterns (B-C vs. B-H), respectively. A total of 895 and 1292 genes were upregulated and downregulated in the N heterogeneous nutrient environment compared with the homogeneous nutrient environment (A-N vs. A-T), accounting for 40.92 and 59.08% of the total DEGs, respectively. The P heterogeneous nutrient environment contained 827 upregulated genes and 1330 downregulated genes compared with the homogeneous nutrient environment (A-P vs. A-T), accounting for 38.34 and 61.66% of all DEGs. 

Furthermore, the K heterogeneous nutrient environment had 1484 (39.00%) upregulated and 2321 (61.00%) downregulated genes relative to the homogeneous nutrient environment (A-K vs. A-T). The unmixed planting and single-plant planting patterns (B-C vs. B-D) contained 857 (54.34%) upregulated and 720 (45.66%) downregulated genes. The upregulated (1750) and downregulated (2037) genes between the mixed and single-plant planting patterns (B-H vs. B-D) accounted for 46.21 and 53.79% of the total DEGs. Furthermore, the pure plant pattern and mixed plant patterns (B-C vs. B-H) had 839 (46.79%) upregulated genes and 954 (53.21%) downregulated genes.

The GO functional classification (Figure 9) showed that the DEGs enriched 27.30, 58.77, and 13.93% of biological processes (BPs), molecular functions (MFs), and cell compositions (CCs), respectively, between the N heterogeneous and homogeneous nutrient environments (A-N vs. A-T). The P heterogeneous and homogeneous nutrient environments (A-P vs. A-T) had 39.30, 54.73, and 5.97% DEGs enriching BPs, MFs, and CCs, respectively. Meanwhile, the DEGs in the K heterogeneous and homogeneous nutrient environments (A-K vs. A-T) enriched 30.98, 54.31, and 14.70% of the BPs, MFs, and CCs, respectively.

In the BPs, A-N vs. A-T were relatively similar to A-P vs. A-T, with DEGs mainly enriching the response to biotic stimulus, response to other organisms, and defense response to other organisms. The DEGs between A-K and A-T mainly enriched DNA geometric change, DNA duplex unwinding, and cellular carbohydrate metabolic processes. In the MF, A-N vs. A-T and A-P vs. A-T were relatively similar, with DEGs mainly enriching in terms such as DNA binding, transcription factor activity, transcription regulator activity, and sequence-specific DNA binding. Under CC, the DEGs of A-N vs. A-T, A-P vs. A-T, and A-K vs. A-T were relatively similar, mainly enriching the cell wall, external encapsulating structure, and apoplast pathways. The DEGs between A-K and A-T mainly enriched two pathways: hydrolase activity and antioxidant activity.

GO functional enrichment analysis of the *C. hodginsii* root tissues in the three planting patterns identified the three most significant terms (Figure 10). Thus, 27.22, 61.24, and 11.54% of the DEGs enriched the BPs, MFs, and CCs of the conspecific neighbor and single-planting patterns (B-C vs. B-D), respectively. Furthermore, the DEGs from the mixed and single-plant patterns (B-H vs. B-D) enriched 27.98, 65.34, and 6.70% of the BPs, MFs, and CCs, respectively. The DEGs from conspecific neighbor and heterospecific neighbor (B-C vs. B-H) enriched 30.28, 24.52, and 45.20% of the BPs, MFs, and CCs, respectively.

The most significant differences in the BP were among the *C. hodginsii* root systems under the three planting patterns. The DEGs of B-C vs. B-D were mainly enriched in response to oxidative stress, response to stress, and cellular ion homeostasis. The DEGs between B-H and B-D mainly enriched cell or subcellular component movement and microtubule-based movement. The DEGs between B-C and B-H mainly enriched the L-phenylalanine biosynthetic and metabolic processes and the peptide metabolic process. Under MF, the DEGs between B-C and B-D mainly enriched oxidoreductase activity, antioxidant activity, and peroxidase activity. However, DEGs between B-H and B-D mainly enriched serine-type peptidase activity, serine hydrolase activity, and serine-type endopeptidase activity. The DEGs between B-C and B-H mainly enriched endopeptidase inhibitor activity, peptidase inhibitor, and regulator activity. In the CC, the DEGs of B-C vs. B-D mainly enriched cell walls, external encapsulating structures, and apoplast pathways. The DEGs between B-H and B-D mainly enriched the chromosome, nucleosome, and DNA packaging complex pathways. Finally, the DEGs between B-C and B-H mainly enriched the ribonucleoprotein complex, ribosomes, and non-membrane-bound organelles (Figure 11). 

KEGG pathway enrichment analysis of the DEGs in *C. hodginsii* root systems under different heterogeneous and homogeneous nutrient environments identified 20 enriched KEGG pathways (Figure 11). These pathways included 308 DEGs (91 upregulated and 217 downregulated) from the N heterogeneous and homogeneous nutrient environments (A-N vs. A-T). The genes that enriched the plant hormone signal transduction pathway showed the highest expression levels, followed by the MAPK signaling pathway’s plant and carbon metabolism genes. The P heterogeneous and homogeneous nutrient environments (A-P vs. A-T) had 273 DEGs (66 upregulated and 207 downregulated). The highest expression of these genes was observed in phenylpropanoid biosynthesis, followed by those related to plant hormone signal transfer and plant–pathogen interaction. Additionally, the K heterogeneous and homogeneous nutrient environments (A-K vs. A-T) contained 507 DEGs (136 upregulated and 371 downregulated). The highest expression was registered in the genes that enriched phenylpropanoid biosynthesis, genes enriching plant hormone signal translation, and the biosynthesis of amino acids (Figure 12).

KEGG pathway enrichment using the DEGs from the roots of *C. hodginsii* under competition from adjacent plants enriched 20 KEGG pathways. Thus, 235 DEGs (141 upregulated and 94 downregulated) between the conspecific neighbor and single-planting patterns (B-C vs. B-D) enriched genes related to phenylpropanoid biosynthesis, followed by the biosynthesis of various plant secondary metabolites and plant hormone signal transfer. The heterospecific neighbor and single-planting patterns (B-H vs. B-D) contained 301 DEGs (141 upregulated and 160 downregulated) that enriched genes related to phenonoid biosynthesis, followed by flavonoid biosynthesis and the biosynthesis of variant plant secondary metabolites. Finally, the conspecific neighbor and heterospecific neighbor (B-C vs. B-H) had 256 DEGs (109 upregulated and 147 downregulated) that enriched spliceosomes, followed by plant hormone signal transduction and phenylpropanoid biosynthesis (Figure 13).

Conclusively, 12 common genes expressed in the root system of *C. hodginsii* under different treatments were used to validate the transcriptome data (Figure 14). The expression of all 12 genes was consistent with those of the transcriptome data, indicating the reliability of the transcriptome sequencing data.

## 4. Discussion

Due to the highly heterogeneous distribution of nutrients in forest soil in geographic space, plants constantly change their root morphology and physiological characteristics to adapt to different heterogeneous nutrient environments during growth. Thus, different plant roots exhibit significantly different characteristics when facing heterogeneous nutrient environments. Trees also face competition from adjacent tree species for space and nutrients. Therefore, the plant growth and root development trends in heterogeneous nutrient environments and adjacent plants significantly differ [24]. In this study, the root vitality of most *C. hodginsii* seedlings in N and P heterogeneous nutrient environments was higher than that in K heterogeneous and homogeneous nutrient environments. Root vitality is an important indicator of plant growth, reflecting the strength of root metabolism. Higher root vitality indicates a stronger ability to absorb nutrients [25,26,27]. The root vitality of most lines in the K heterogeneous environment was lower than that in the homogeneous nutrient environment, indicating that N and P heterogeneous nutrient environments promote root vitality in most *C. hodginsii* families compared to in homogeneous nutrient environments. The K heterogeneous environment unbalanced the K ion concentration in *C. hodginsii* seedlings, affecting its metabolism and inhibiting the root vitality of *C. hodginsii*.

In this study, the root vitality of all *C. hodginsii* lines under unmixed and mixed planting patterns was higher than that under the single-plant planting pattern, with some differences reaching a significant level. However, the root vitality of all lines under the mixed planting pattern was higher than that under the pure planting pattern. In mixed-planted ryegrass (*Lolium perenne*) and *Poa pratensis*, the growth rate and root vitality of ryegrass in nutrient patches are significantly higher than those of the single-plant planting pattern, consistent with the results of this study [17,28]. Therefore, plants in a competitive environment enhance their root vitality by accelerating the root metabolism intensity, enhancing their competitiveness in soil resources, and increasing their ability to absorb nutrients.

In this study, the average CAT activity in the root system of most *C. hodginsii* seedlings peaked in the P heterogeneous nutrient environment. Superoxide dismutase (SOD), POD, and CAT mainly protect plants from external stress and help plants avoid the damage of reactive oxygen species to plant cells. They can effectively improve the adaptability and resistance of plants to environmental stresses, and their activity is often used as a physiological and biochemical indicator of plant aging [29,30]. In the N heterogeneous nutrient environment, the average CAT activity in the root system of each family was higher than that in the homogeneous environment but slightly lower than that in the P heterogeneous nutrient environment. 

In the K heterogeneous nutrient environment, the average CAT activity in most lines was slightly higher than in the homogeneous environment, but the difference was insignificant. The POD activity in the root system showed a similar trend to CAT activity. Thus, the average SOD activity in the root system was higher in most families with N, P, and K heterogeneous than in homogeneous nutrient environments, as in a previous study [14]. 

Nitrogen and P are highly demanded nutrients by plants. When encountering soil N and P heterogeneous environments, some plant roots in nutrient-poor patches may obtain insufficient nutrients, possibly stimulating plant transmission of induction signals. Furthermore, this condition promotes root growth and improves antioxidant enzyme activity in roots to meet metabolic activities. The K heterogeneous nutrient environment showed no significant difference in the antioxidant enzyme activity in the *C. hodginsii* root system compared to the homogeneous environment. The lower tolerance of *C. hodginsii* to the K heterogeneous nutrient environment may explain this scenario because it inhibits the antioxidant enzyme activity, lowering antioxidant enzyme activity compared to N and P heterogeneous nutrient environments. The MDA content in the roots of *C. hodginsii* seedlings in N and P heterogeneous nutrient environments showed an unclear pattern compared to in the homogeneous nutrient environment. However, most families had higher MDA content in the K heterogeneous nutrient environment roots than in other nutrient environments. Therefore, *C. hodginsii* has a low tolerance to a K heterogeneous nutrient environment, significantly damaging the cells and adaption to heterogeneous environments.

The average values of CAT and POD activities in *C. hodginsii* seedlings peaked in the mixed planting pattern. The average values of CAT and POD activities in the unmixed planting pattern were also higher than those in the single-plant planting pattern, indicating that competition patterns promoted the CAT and POD activities in the root systems of various *C. hodginsii* lines. Therefore, the intensity and form of competition among adjacent plants partly determine CAT and POD activities. As with root biomass and vitality, when *C. hodginsii* roots encounter competition from adjacent plants during the foraging process, the root metabolism increases to achieve higher competitiveness. Increased metabolism produces excessive hydrogen peroxide, damaging the plant. Therefore, the hydrogen peroxide must be rapidly converted into other harmless or less toxic substances to avoid such damage. Cells often use CAT to catalyze hydrogen peroxide decomposition, while POD has a dual effect of eliminating hydrogen peroxide, phenol, and amine toxicity. Therefore, under competitive patterns, the CAT and POD activities in the root system of *C. hodginsii* would significantly increase to eliminate the harmful substances produced by increased metabolism [31]. 

The average root SOD activity of all lines in the mixed planting pattern was higher than that in the single-plant planting and unmixed planting patterns. In the unmixed planting pattern, most families were slightly higher than those in the single-plant planting pattern, but the difference was insignificant. This reaction may be due to the low CAT and POD activities during single-plant planting, resulting in less H_2_O_2_ production. Furthermore, plants need to increase SOD activity and catalyze the dismutation of superoxide anions, generating H_2_O_2_ and O_2_ to maintain low free radicals in the cells and avoid the cell membrane damage caused by free radicals [32]. The average MDA content in the roots of most lines in the unmixed planting pattern was higher than that in the single-plant and mixed planting patterns. 

In contrast, the average MDA content in each family in the single-plant planting pattern was slightly lower than that in the mixed planting pattern, but the difference was insignificant. This response indicates that intraspecific competition can significantly increase the MDA content in the roots of *C. hodginsii* family seedlings more than non-competitive and interspecific competition. Lipid peroxidation of the plant membrane produces MDA, and its content can reflect the degree of damage to the cell membrane system [33]. Competition from adjacent plants causes nutrient and spatial resource deficiencies due to their demands and ecological niches, increasing the MDA content in the root system. Adjacent plant competition from heterologous non-dominant tree species can effectively increase soil nutrient utilization and spatial resource efficiency. Finally, stress levels drop, decreasing the MDA content in the root system.

The ability to absorb and utilize N and P varies among different plant species and varieties [34]. Liang et al. showed that P-efficient Chinese fir clones have a higher ability to absorb and utilize P. At the same time, there is no significant correlation between P absorption and the utilization efficiency of high-efficiency Chinese fir [35]. In this study, the average contents of N and P in the roots of most families peaked in a P heterogeneous nutrient environment, possibly because the P heterogeneous nutrient environment caused the *C. hodginsii* roots to colonize nutrient-rich patches. This pattern promoted the absorption and utilization of N and P by the roots to accumulate more biomass and meet the requirements of colonization growth. The *C. hodginsii* root system in the N heterogeneous nutrient environment had a higher N and P content than the homogeneous nutrient environment but lower than in the P heterogeneous nutrient environment. This higher N and P content may be related to the function of P in the soil. 

Plants mainly rely on root proliferation and colonization to increase soil P acquisition and improve the efficiency of P absorption, utilization, and transformation [3]. The N heterogeneous environment can also induce lateral proliferation of the *C. hodginsii* root to absorb soil nutrients and improve nutrient patch utilization. However, since N is more easily diffused in the soil than P when acquiring N in patches, the root system of *C. hodginsii* does not need to proliferate as much as when acquiring P in patches. Therefore, under a P heterogeneous nutrient environment, the biomass and the efficiency of nutrient absorption of the *C. hodginsii* root system would be higher than in the N heterogeneous nutrient environment. The N and P contents in the roots of most *C. hodginsii* families were significantly higher than those in K heterogeneous and homogeneous nutrient environments, similar to the conclusion of Pang et al. [36]. The N or P heterogeneous nutrient condition improved the root proliferation ability of *C. hodginsii* seedlings, enhancing the absorption and utilization of soil P by *C. hodginsii* seedlings and promoting the growth of *C. hodginsii* seedlings.

In this study, the root N and P content of most *C. hodginsii* families was higher in the competitive patterns than in the non-competitive pattern, and the mixed planting pattern had a higher promoting effect on the N and P accumulation in the roots of most *C. hodginsii* lines than in the unmixed planting pattern. This response indicates that the competitive pattern can effectively promote N and P accumulation in the roots of most *C. hodginsii* lines. Thus, the promoting effect of interspecific competition is higher than that of intraspecific competition, possibly explaining the hypothesis that adjacent plant competition can induce changes in plant behavior, including root growth and biomass accumulation [37,38]. 

Plants promote nutrient absorption and accumulation in the roots to meet the demands of root colonization and growth. The interaction varies between the roots of the same and different plant species [39,40]. Many studies attribute this phenomenon to discrimination between self and non-self roots. When plant roots encounter non-self roots, they produce more roots, promoting nutrient absorption efficiency from the soil and increasing root nutrient accumulation. 

Carbon is the fundamental element that constitutes the structure of plants, it is the source of energy for various life activities, and it plays a crucial role in the growth and development of plants [41]. This study showed that the average C content in the roots of most *C. hodginsii* lines peaked in a P heterogeneous nutrient environment. The average C content in the roots of the N heterogeneous nutrient environment was generally higher than that in the K heterogeneous nutrient environment. However, both overall averages were higher than those in a homogeneous nutrient environment, indicating that a heterogeneous nutrient environment is conducive for C accumulation in *C. hodginsii* root systems. This reaction may be because C is the basic element of the root system, which is important for root lignification to enhance root structure and colonization. 

A heterogeneous nutrient environment is conducive to the colonization of *C. hodginsii* root systems, causing higher C contents in the *C. hodginsii* root system than in the homogeneous environment. The P heterogeneous nutrient environment had the best promotion effect on *C. hodginsii* root system colonization, leading to higher C contents in its root. The average C content in most roots peaked in the mixed planting pattern, indicating that the mixed planting pattern can improve C accumulation in the roots.

This study conducted a transcriptome analysis of *C. hodginsii* root systems under seven different heterogeneous nutrient environments and adjacent plant competition treatments. Combining the existing *C. hodginsii* genome sequences with high-throughput sequencing can be used to explore the transcriptomic differences in *C. hodginsii* root systems under different treatments. The raw data obtained from sequencing were filtered to remove low-quality sequences, adaptors, and reads with a high content of unknown base N that may affect subsequent experimental analysis [42]. The samples had 43.69–54.12 Mb of clean reads, accounting for >95% of the total reads. These data are similar to the transcriptome sequencing results of Zhou et al. for identifying and analyzing the R2R3-MYB transcription factors in *C. hodginsii* [43]. 

A total of 3805 differentially expressed genes were screened out in the K heterogeneous nutrient environment and the homogeneous nutrient environment (A-K vs. A-T), which was much higher than the number of differentially expressed genes in the N and P heterogeneous nutrient environments and the homogeneous nutrient environment. According to the results of GO enrichment analysis, A-N vs. A-T and A-P vs. A-T were similar. The differentially expressed genes were mainly concentrated in DNA binding transcription factor activity, transcription regulator activity, sequence-specific DNA binding, and terpenoid synthase activity, while A-K vs. A-T differentially expressed genes were mainly concentrated in hydrolase activity, antioxidant enzyme activity, and sequence-specific DNA binding. This is similar to the results of Li [44]. This may be because the K heterogeneous nutrient environment is more unfavorable to the growth and biomass accumulation of *C. hodginsii* than the N, P heterogeneous, and homogeneous nutrient environments. The relative evaluation of phenotypic traits and physiological indicators is also poor, which also requires more tissue physiological activities to participate in the process of *C. hodginsii* adapting to K heterogeneous nutrients, resulting in significant differences in expression genes from other nutrient environments. From the results of the study on the root activity of *C. hodginsii* seedlings in different nutrient environments, it can be seen that the root activity and antioxidant enzyme activity in the K heterogeneous nutrient environment are significantly lower than those in the N and P heterogeneous nutrient environments and the homogeneous environment. Compared with other nutrient environments, the differential genes are mainly focused on the expression of antioxidant enzyme activities; however, the difference in antioxidant enzyme activities between the N and P heterogeneous nutrient environments was small, so the number of differentially expressed genes was small.

Compared with the single plant, the differences in the root transcriptome genes of *C. hodginsii* in the conspecific neighbor and heterospecific neighbor were mainly concentrated in molecular function. The B-C vs. B-D differential genes were mainly enriched in oxidoreductase activity, antioxidant activity, peroxidase activity, etc. B-H vs. B-D differential genes were mainly enriched in serine peptidase activity, serine hydrolase activity, serine endopeptidase activity, etc. This may be because the competitive mode will induce plant roots to increase the intensity of physiological activity. This makes the fine-root enzyme activity of *C. hodginsii* in the competitive mode significantly increased compared with the single-planting mode so that the roots have strong colonization and foraging ability in the competitive environment to obtain a competitive advantage [45]. The differences in the transcriptome genes of the roots of *C. hodginsii* in the conspecific neighbor and heterospecific neighbor were more concentrated in cell composition. The B-C vs. B-H differential genes were mainly enriched in the ribonucleoprotein complex, ribosomes, non-membranous organelles, etc.

The results of KEGG enrichment analysis showed that the number of differentially expressed genes in the roots of *C. hodginsii* in N, P, and K heterogeneous nutrient environments was 308, 207, and 507, respectively, compared with homogeneous nutrient environments. Among them, the expression levels of plant hormone signal transduction and phenylpropanone biosynthesis were generally high. It can be speculated that the difference in nutrient environment will make the roots change the synthesis of their own metabolites through hormone signals so as to better adapt to environmental changes, which is similar to the study of Gan [46].

## 5. Conclusions

The root activity, antioxidant enzyme activity, and element content of *C. hodginsii* seedlings in N and P heterogeneous nutrient environments were higher than those in K heterogeneous and homogeneous nutrient environments. Due to the low tolerance of *C. hodginsii* to the K heterogeneous nutrient environment, the root activity of seedlings in the K heterogeneous environment was lower than that in the homogeneous environment. The root activity, antioxidant enzyme activity, and element contents of *C. hodginsii* under the conspecific neighbor and heterospecific neighbor were generally higher than those under the single-planting mode, while those under the heterospecific neighbor were higher than those under the conspecific neighbor. There were significant or extremely significant interactions among planting pattern, nutrient environment, and family differences for the root activity, CAT activity, POD activity, and MDA content of *C. hodginsii*. The families with higher root physiological plasticity index of *C. hodginsii* in each family were lines 490, 535, and 539. The comprehensive evaluation of root physiological indexes of line 539 was up to 0.806. The model with the highest comprehensive evaluation score of principal component analysis was P heterogeneous nutrient environment × heterospecific neighbor.

This study failed to further explain the adaptability of *C. hodginsii* seedlings to different concentrations of heterogeneous nutrient environments from the perspective of nutrient heterogeneity at different concentrations. It can further explore the response differences of *C. hodginsii* families to different concentrations of heterogeneity in different heterogeneous nutrient environments, which is conducive to improving the theoretical basis of *C. hodginsii* ‘s adaptation mechanism to heterogeneous nutrient environments.

## Figures and Tables

**Figure 1 plants-13-02641-f001:**
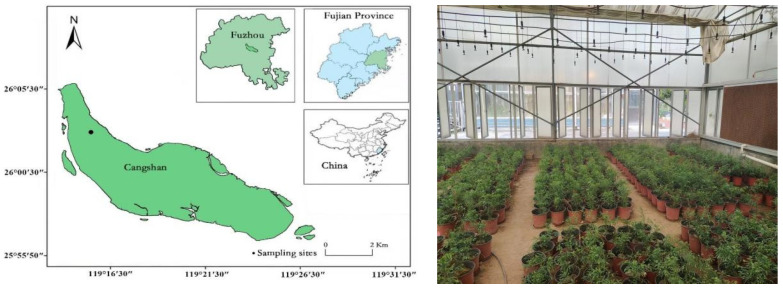
Geographical location of the greenhouses in Fujian Agriculture and Forestry University. (Note: The blue part represents Fujian Province, the light green represents Fuzhou City, and the dark green represents Cangshan District.)

**Figure 2 plants-13-02641-f002:**
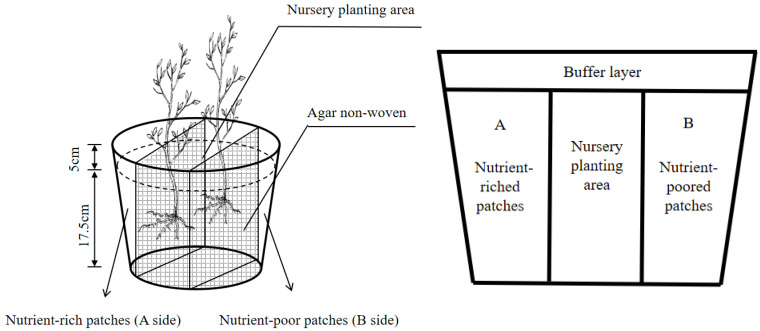
A front view of the pot container.

**Figure 3 plants-13-02641-f003:**
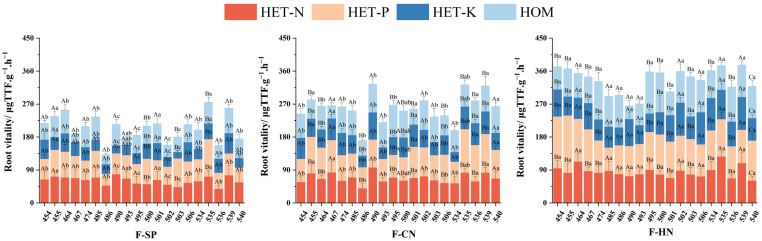
Differences in the root system activities of *C. hodginsii* family seedlings under different treatments. Note: Different uppercase letters represent significant differences in the indices of *C. hodginsii* seedlings in different nutrient environments under the same planting pattern (*p* < 0.05). Different lowercase letters represent significant differences in the indices of *C. hodginsii* seedlings in different planting patterns under the same nutrient environments (*p* < 0.05). HET-N, N heterogeneous nutrient environment; HET-P, P heterogeneous nutrient environment; HET-K, K heterogeneous nutrient environment; HOM, homogeneous environment; F-SP, single-plant pattern; F-CN, conspecific neighbor; F-MP, heterospecific neighbor. Error lines represent standard errors.

**Figure 4 plants-13-02641-f004:**
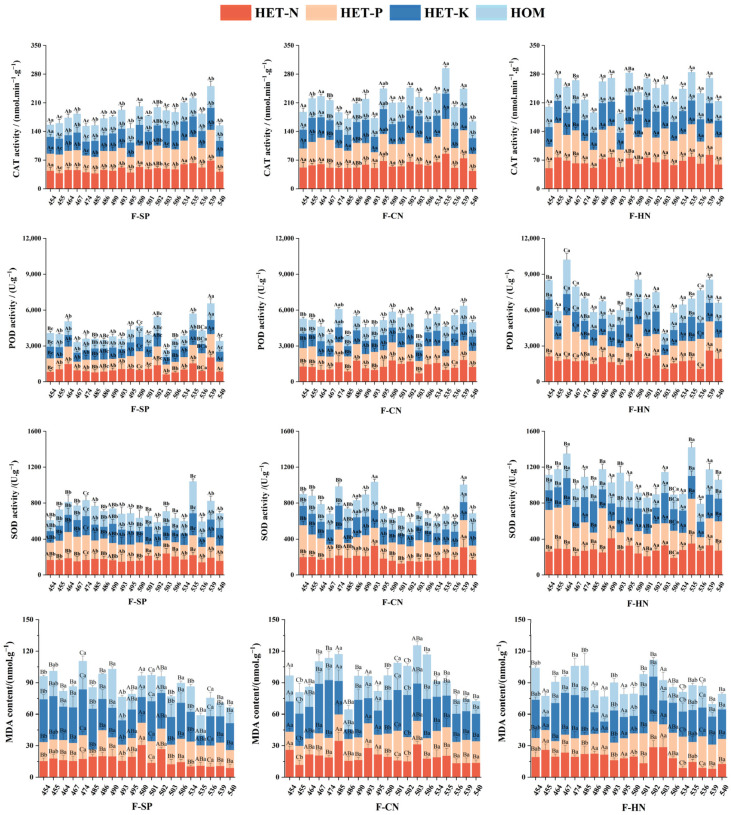
Differences in antioxidant enzyme activities and MDA contents in the root systems of *C. hodginsii* seedlings under different treatments. Note: Different uppercase letters represent significant differences in the indices of *C. hodginsii* seedlings in different nutrient environments under the same planting pattern (*p* < 0.05). Different lowercase letters represent significant differences in the indices of *C. hodginsii* seedlings in different planting patterns under the same nutrient environments (*p* < 0.05). HET-N, N heterogeneous nutrient environment; HET-P, P heterogeneous nutrient environment; HET-K, K heterogeneous nutrient environment; HOM, homogeneous environment; F-SP, single-plant pattern; F-CN, conspecific neighbor; F-MP, heterospecific neighbor. Error lines represent standard errors.

**Figure 5 plants-13-02641-f005:**
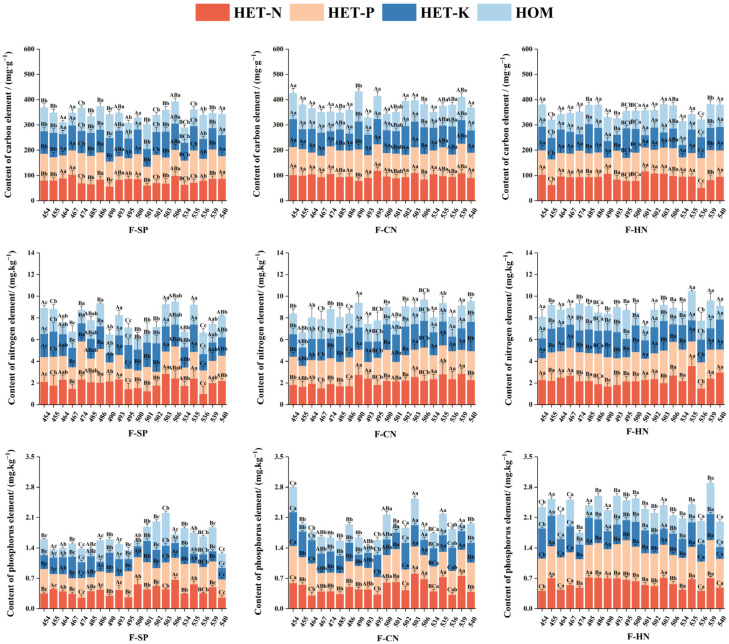
Nutrient content differences in the root system of *C. hodginsii* seedlings under different treatments. Note: Different uppercase letters represent significant differences in the indices of *C. hodginsii* seedlings in different nutrient environments under the same planting pattern (*p* < 0.05). Different lowercase letters represent significant differences in the indices of *C. hodginsii* seedlings in different planting patterns under the same nutrient environments (*p* < 0.05). HET-N, N heterogeneous nutrient environment; HET-P, P heterogeneous nutrient environment; HET-K, K heterogeneous nutrient environment; HOM, homogeneous environment; F-SP, single-plant pattern; F-CN, conspecific neighbor; F-MP, heterospecific neighbor. Error lines represent standard errors.

**Figure 6 plants-13-02641-f006:**
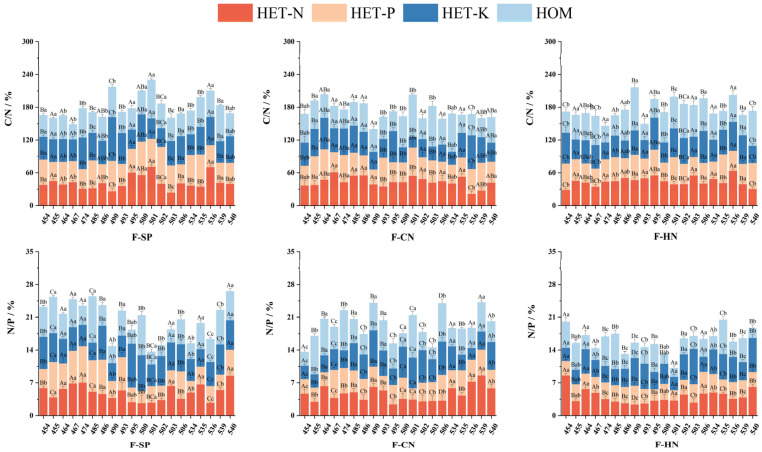
Elemental stoichiometry characteristics of the root system of *C. hodginsii* seedlings under different treatments. Note: Different uppercase letters represent significant differences in the indices of *C. hodginsii* seedlings in different nutrient environments under the same planting pattern (*p* < 0.05). Different lowercase letters represent significant differences in the indices of *C. hodginsii* seedlings in different planting patterns under the same nutrient environments (*p* < 0.05). HET-N, N heterogeneous nutrient environment; HET-P, P heterogeneous nutrient environment; HET-K, K heterogeneous nutrient environment; HOM, homogeneous environment; F-SP, single-plant pattern; F-CN, conspecific neighbor; F-MP, heterospecific neighbor. Error lines represent standard errors.

**Figure 7 plants-13-02641-f007:**
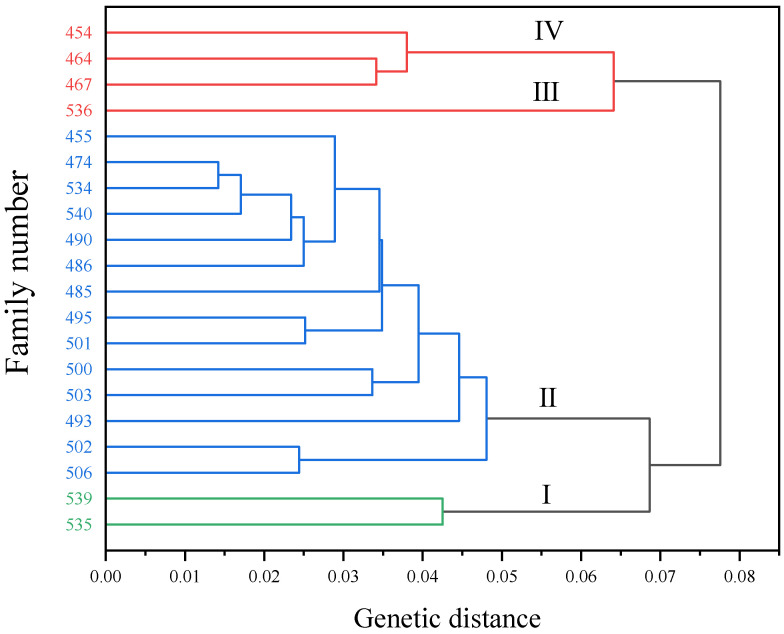
Cluster analysis of *C. hodginsii* families. Note: In the figure, the green part represents group I, the blue represents group II, and the red represents group III and group IV.

**Figure 8 plants-13-02641-f008:**
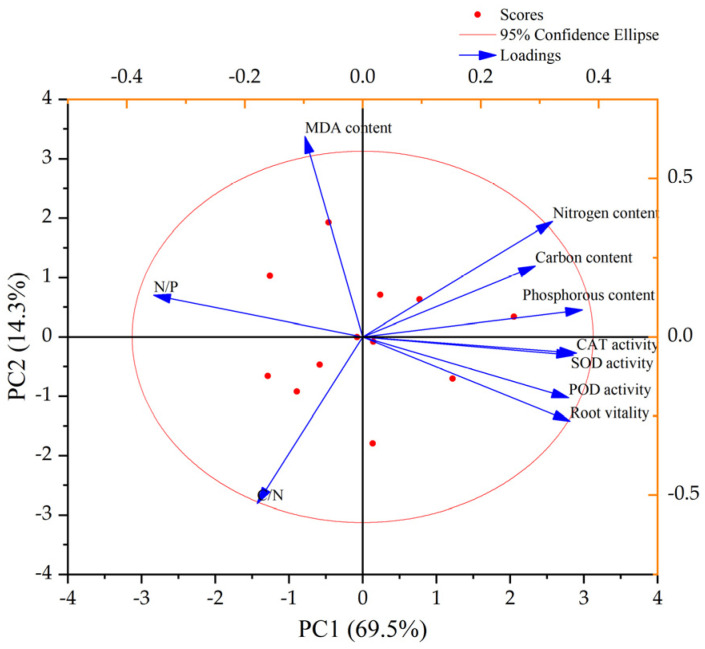
Principal component analysis plot of physiological indicators of the *C. hodginsii* root system under different treatments.

**Figure 9 plants-13-02641-f009:**
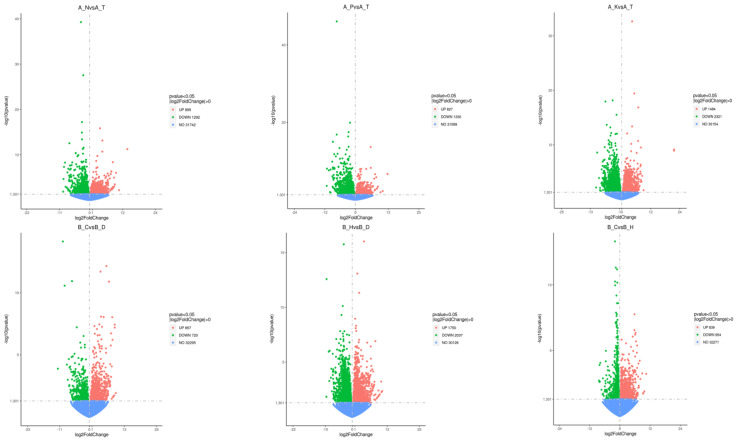
Distribution of differentially expressed genes in different comparison combinations.

**Figure 10 plants-13-02641-f010:**
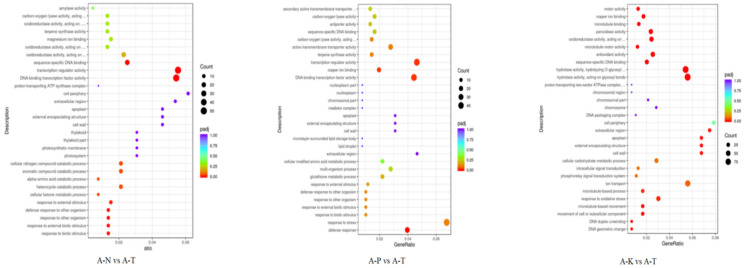
The GO bubble plot of the *C. hodginsii* root system in a heterogeneous nutrient environment. Note: The horizontal axis in the figure represents the ratio of DEGs annotated in the GO term to the total number of DEGs, while the vertical axis represents the GO term. The size of the dots represents the number of genes annotated in the GO term, and the red to purple color range represents the significance of the enrichment. The same applies below.

**Figure 11 plants-13-02641-f011:**
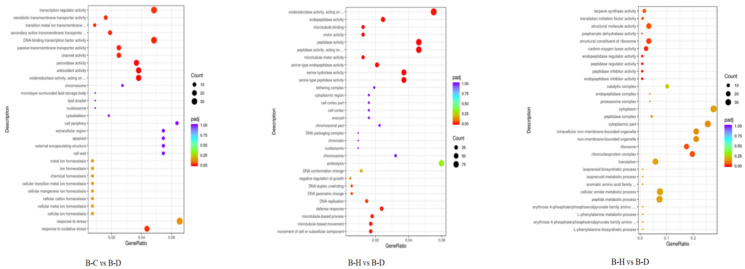
The GO bubble plot of *C. hodginsii* in different planting patterns.

**Figure 12 plants-13-02641-f012:**
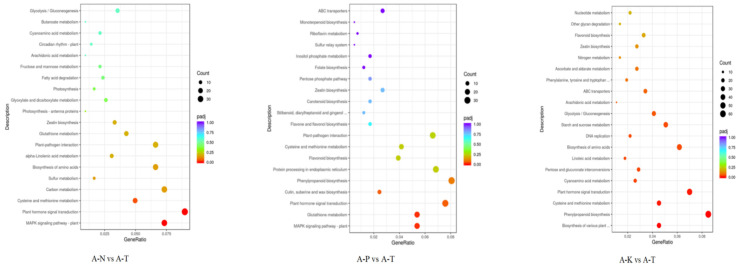
The KEGG bubble plot of *C. hodginsii* under a heterogeneous nutrient environment.

**Figure 13 plants-13-02641-f013:**
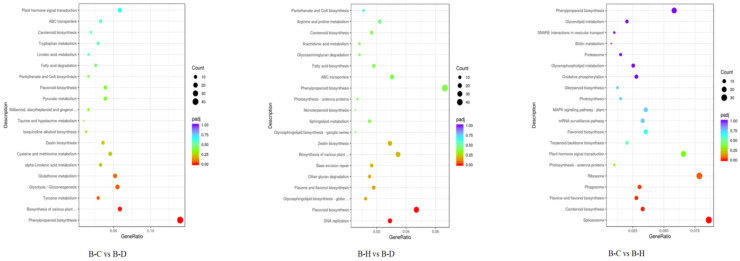
The KEGG bubble plot of *C. hodginsii* under planting patterns.

**Figure 14 plants-13-02641-f014:**
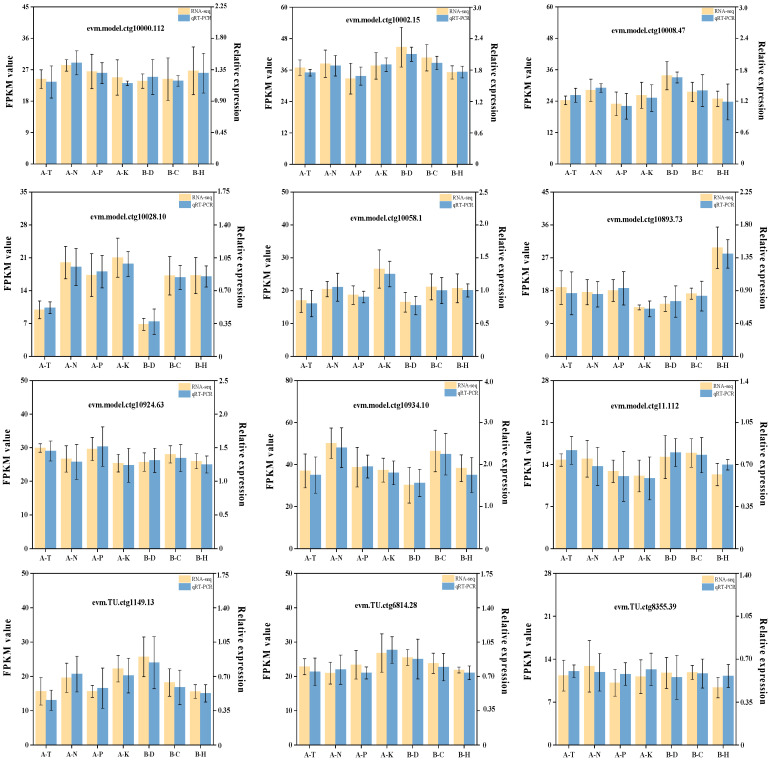
qRT PCR validation of 12 candidate genes from the transcriptome of *C. hodginsii* roots.

**Table 1 plants-13-02641-t001:** Line numbers and sources of different *C. hodginsii* families.

Seedling	Line Number	Germplasm Number (of a Species)	Source
1	454	HNSN-A2 view	Hunan Suining Zhai Mayor Puzi Village
2	455	HNSN-B1	Hunan Suining Zhai Mayor Puzi Village
3	464	GDNL-24	Xiaohuangshan Mountain Climbing Road, Nanyuan County, Shaoguan City, Guangdong Province, China
4	467	Guangxi 3	Guangxi Ecological Engineering Vocational and Technical College
5	474	HNCS B-3	Tangtian Forestry, Ansha Town, Changsha, Hunan
6	485	HNCS A-3	Tangtian Forestry, Ansha Town, Changsha, Hunan
7	486	HNCS A-2	Tangtian Forestry, Ansha Town, Changsha, Hunan
8	490	GDLK-3	Yao Village, Luokeng Town, Qujiang District, Shaoguan, Guangdong, China
9	493	Guangxi Yulin Yushu 5	Guangxi Yulin Liwan Forestry
10	495	NDFK-2	Ningde Fukou Forestry 2
11	500	NDFK-7	Ningde Fukou Forestry 7
12	501	NDFK-8	Ningde Fukou Forestry 8
13	502	NDFK-9	Ningde Fukou Forestry 9
14	503	NDFK-10	Ningde Fukou Forestry 10
15	506	NDFK-13	Ningde Fukou Forestry 13
16	534	LYBS-2	Longyan Shanghang Baisha Forestry No. 2
17	535	LYBS-3	Longyan Shanghang Baisha Forestry No. 3
18	536	LYBS-4	Longyan Shanghang Baisha Forestry No. 4
19	539	LYBS-7	No. 7, Baisha Forestry, Shanghang, Longyan
20	540	LYBS-8	No. 8, Baisha Forestry, Shanghang, Longyan

**Table 2 plants-13-02641-t002:** Concentrations of N, P, and K in the heterogeneous (HET) and homogeneous nutrient patches [2,17,18] (mg/kg).

Nutrient Patch	Heterogeneous Nutrient Patch						Homogeneous Nutrient Patch					
	Nutrient-Rich Patches (A Side)			Nutrient-Poor Patches (B Side)			A Side			B Side		
	N	P	K	N	P	K	N	P	K	N	P	K
Heterogeneous plaque N (HET-N)	100	125	75	0	125	75	50	125	75	50	125	75
P heterogeneous plaque (HET-P)	50	250	75	50	0	75						
Heterogeneous plaque K (HET-K)	50	125	150	50	125	0						

**Table 3 plants-13-02641-t003:** Primer pairs designed for real-time fluorescence quantitative PCR.

Primer Name	Forward Primer	Reverse Primer
evm.model.ctg10000.112	GTGCCAATTGTGGACCTCCT	CAACCTGGCCACTCTCCAAA
evm.model.ctg10002.15	TCCAAGAGAGGTGTGGCTCT	ACCTTGTTCACTTGGGCCTC
evm.model.ctg10008.47.1	AGTTCCGTCTTGCTGATGGC	TGGCCACTGACCAATTATGCT
evm.model.ctg10028.10	TCAGGCATTCTGCAAGGACG	TTTGCACAGCCACGAACTGA
evm.model.ctg10058.1	GAGCAAAGCGACACAGGAAG	ACCTCCTCTCCTGGCAAGAC
evm.model.ctg10893.73	TATTCCCAGCACTTGGCGTT	TGCCTGATCCGCTGACATTT
evm.model.ctg10924.63	CTGTGGAAGCGTACTGGAGG	TCGCAAATGGACTGGAACGA
evm.model.ctg10934.10	CGACTGTGCTCTGCTCTTGA	TTGACGCTCGGCTTAACCAT
evm.model.ctg11.112	TGCAGTGGTCTTCTGGTTGG	GAGTCCCGCTTATTACGCCT
evm.model.ctg1149.13	CCCTCGAGCTATACCAGCAC	ACGTTCTATGGACTCGCCAC
evm.model.ctg6814.28	ATCAACAGCAGGTTTCGCCA	TCAAGTTGAGCGTGAGGGAG
evm.model.ctg8355.39	GCAACCCGACCTTAGGAGTG	GCACATACTCCCATGCTCCA

**Table 4 plants-13-02641-t004:** Effects of different treatments and families on root physiological traits of *C. hodginsii*.

Factor	*F* Value of Each Index									
	Root Vitality	CAT Activity	POD Activity	SOD Activity	MDA Content	Carbon Content	Nitrogen Content	Phosphorous Content	C/N	N/P
Planting pattern (a)	10.968 **	12.546 **	14.723 **	12.918 **	2.248 ^ns^	7.067 **	7.408 **	18.684 **	0.615 ^ns^	10.892 **
Nutrient environment (b)	11.387 **	13.836 **	10.366 **	10.784 **	12.684 **	22.759 **	24.535 **	21.235 **	1.998 *	3.483 ^ns^
Family (c)	22.645 **	44.425 **	32.645 **	16.239 **	33.568 **	6.794 *	11.235 **	37.457 **	49.564 **	28.454 **
a × b	4.536 *	0.785 ^ns^	0.649 ^ns^	1.190 ^ns^	5.337 *	2.546 *	0.233 ^ns^	2.126 *	1.154 ^ns^	0.643 ^ns^
a × c	6.004 *	3.565 *	10.568 **	0.793 ^ns^	5.687 *	4.985 *	6.952 *	1.003 ^ns^	0.078 ^ns^	0.278 ^ns^
b × c	12.385 **	9.687 **	11.567 **	4.955 *	1.335 ^ns^	9.432 **	6.003 *	0.779 ^ns^	0.136 ^ns^	0.995 ^ns^
a × b × c	4.368 *	3.684 *	5.664 *	0.689 ^ns^	3.006 *	1.324 ^ns^	0.789 ^ns^	0.146 ^ns^	0.384 ^ns^	0.779 ^ns^

*, *p* < 0.05; **, *p* < 0.01; ^ns^, *p* > 0.05.

**Table 5 plants-13-02641-t005:** Physiological plasticity index of root growth activity of *C. hodginsii* families.

Family Number	Indicators										
	Root Vitality	CAT Activity	POD Activity	SOD Activity	MDA Content	Carbon Content	Nitrogen Content	Phosphorous Content	C/N	N/P	Overall Phenotypic Plasticity Index
454	0.687	0.446	0.632	0.748	0.617	0.279	0.535	0.636	0.489	0.673	0.574
455	0.780	0.561	0.489	0.637	0.712	0.403	0.480	0.768	0.335	0.650	0.581
464	0.575	0.448	0.662	0.706	0.604	0.617	0.566	0.759	0.389	0.621	0.595
467	0.816	0.467	0.708	0.533	0.681	0.520	0.436	0.598	0.552	0.648	0.596
474	0.653	0.460	0.488	0.466	0.682	0.529	0.654	0.660	0.483	0.588	0.566
485	0.665	0.265	0.496	0.478	0.820	0.443	0.344	0.767	0.420	0.694	0.539
486	0.703	0.473	0.664	0.737	0.689	0.379	0.467	0.721	0.426	0.638	0.590
490	0.834	0.476	0.444	0.610	0.712	0.549	0.693	0.685	0.671	0.693	0.637
493	0.783	0.388	0.593	0.482	0.499	0.445	0.372	0.825	0.482	0.636	0.550
495	0.736	0.475	0.612	0.531	0.462	0.483	0.464	0.844	0.447	0.742	0.579
500	0.623	0.316	0.661	0.497	0.478	0.557	0.625	0.840	0.546	0.742	0.589
501	0.545	0.453	0.623	0.523	0.702	0.458	0.486	0.723	0.580	0.614	0.571
502	0.722	0.380	0.590	0.575	0.641	0.417	0.517	0.495	0.509	0.669	0.552
503	0.841	0.434	0.519	0.557	0.653	0.453	0.516	0.551	0.675	0.676	0.587
506	0.451	0.245	0.491	0.576	0.655	0.342	0.516	0.725	0.475	0.604	0.508
534	0.750	0.343	0.497	0.536	0.728	0.528	0.555	0.527	0.660	0.573	0.570
535	0.631	0.348	0.717	0.606	0.702	0.571	0.620	0.569	0.693	0.655	0.611
536	0.677	0.351	0.609	0.638	0.692	0.499	0.359	0.592	0.445	0.580	0.544
539	0.569	0.486	0.555	0.543	0.651	0.580	0.665	0.841	0.746	0.796	0.643
540	0.614	0.433	0.577	0.516	0.683	0.208	0.590	0.625	0.519	0.752	0.552

**Table 6 plants-13-02641-t006:** Comprehensive evaluation of *C. hodginsii* lines based on root physiological indicators.

Family Number	Membership Function Value					Comprehensive Evaluation	Rank
	μ1	μ2	μ3	μ4	μ5		
454	0.739	0.006	0.762	0.693	0.062	0.487	12
455	0.667	0.303	0.590	0.749	0.463	0.555	7
464	0.618	0.127	0.626	0.000	0.415	0.394	17
467	0.323	0.096	0.507	0.561	0.060	0.304	20
474	0.697	0.249	1.000	0.987	0.383	0.645	3
485	0.885	0.384	0.524	0.336	0.956	0.633	5
486	0.587	0.455	0.248	0.333	0.069	0.411	16
490	0.598	0.471	0.596	0.855	1.000	0.642	4
493	0.594	0.000	0.000	1.000	0.312	0.373	18
495	0.300	0.433	0.637	0.823	0.041	0.435	15
500	0.349	0.440	0.758	0.631	0.174	0.462	13
501	0.000	0.738	0.706	0.456	0.827	0.447	14
502	0.390	0.758	0.695	0.742	0.015	0.544	10
503	0.577	0.973	0.121	0.482	0.257	0.554	8
506	0.867	0.725	0.567	0.276	0.124	0.628	6
534	0.731	0.354	0.423	0.708	0.317	0.541	11
535	1.000	0.545	0.690	0.728	0.583	0.757	2
536	0.242	0.223	0.812	0.236	0.307	0.339	19
539	0.916	1.000	0.822	0.736	0.000	0.806	1
540	0.956	0.173	0.745	0.254	0.111	0.547	9

**Table 7 plants-13-02641-t007:** Principal component analysis of the physiological indicators of the *C. hodginsii* root system under different treatments.

Parameters	Principal Component	
	1	2
Root vitality	0.927	−0.318
CAT activity	0.956	−0.061
POD activity	0.921	−0.231
SOD activity	0.942	−0.07
MDA content	−0.258	0.756
Carbon content	0.772	0.268
Nitrogen content	0.849	0.436
Phosphorous content	0.982	0.101
C/N	−0.472	−0.63
N/P	−0.935	0.157
Eigenvalue	6.952	1.428
Contribution rate/%	69.521	14.284
Accumulative contribution/%	69.521	83.805

**Table 8 plants-13-02641-t008:** Physiological indicators of *C. hodginsii* seedlings under different nutrient environments and planting patterns.

Treatments (Planting Pattern × Nutrient Heterogeneous Environments)	Comprehensive Scores	Comprehensive Rank
F-SP × HET-N	−1.15	9
F-SP × HET-P	−0.14	7
F-SP × HET-K	−2.13	11
F-SP × HOM	−2.48	12
F-CN × HET-N	0.25	5
F-CN × HET-P	1.52	3
F-CN × HET-K	−0.52	8
F-CN × HOM	−1.80	10
F-HN × HET-N	2.12	2
F-HN × HET-P	3.82	1
F-HN × HET-K	0.56	4
F-HN × HOM	−0.06	6

Note: HET-N represents N heterogeneous nutrient environment, HET-P indicates P heterogeneous nutrient environment, HET-K represents K heterogeneous nutrient environment, and HOM represents a homogeneous nutrient environment. F-SP indicates single-plant planting (non-competitive pattern), F-CN represents pure *C. hodginsii* planting (conspecific neighbor), and F-HN represents mixed *C. hodginsii*-*C. lanceolata* planting (heterospecific neighbor).

**Table 9 plants-13-02641-t009:** Transcriptome sequencing data quality.

Sample	Raw Reads	Clean Reads	Error Rate (%)	Q20 (%)	Q30 (%)	GC Pct (%)
A-T-1	47918836	46088028	0.03	97.43	92.89	42.31
A-T-2	49063524	46927070	0.03	97.720	93.54	42.6
A-T-3	46409234	44433642	0.03	96.91	91.75	42.36
A-N-1	47116614	44807186	0.03	97.51	93.03	41.69
A-N-2	48275896	46101862	0.03	97.17	92.24	41.91
A-N-3	48952068	45821786	0.03	97.060	92	40.71
A-P-1	47393138	45495802	0.03	97.26	92.49	42.74
A-P-2	56049694	54122788	0.03	97.71	93.61	42.82
A-P-3	46605134	44047126	0.03	96.97	91.98	43.18
A-K-1	49595876	47873076	0.03	97.340	92.72	42.45
A-K-2	47160248	45626714	0.03	97.1	92.19	42.62
A-K-3	52638216	51459180	0.02	98.080	94.16	42.37
B-D-1	47995440	46339908	0.03	97.57	93.02	40.52
B-D-2	50715898	48051502	0.03	97.52	93.04	42.2
B-D-3	48475284	46106702	0.03	97.120	92.15	41.69
B-C-1	46314474	43688356	0.03	97.24	92.42	41.7
B-C-2	47112412	44860262	0.03	97.34	92.59	41.57
B-C-3	49099718	46889444	0.03	96.810	91.52	41.94
B-H-1	48886278	47388762	0.03	96.89	91.77	42.84
B-H-2	50421026	48131726	0.03	97.87	93.93	43.29
B-H-3	49052546	46943376	0.03	97.08	92.21	42.91

Note: A-T represents the homogeneous nutrient environment, A-N represents the N heterogeneous nutrient environment, A-P represents the P heterogeneous nutrient environment, and A-K represents the K heterogeneous nutrient environment; B-D represents the single-plant planting pattern, B-C represents the P planting pattern, and B-H represents the mixed planting pattern. Q20 and Q30 indicate the proportion of filtered data having bases with <1% and <0.1% error rates compared to the original data; GC represents the proportion of G and C among the four bases in the clean reads.

**Table 10 plants-13-02641-t010:** Statistics on the mapping rate to the reference genomes.

Sample	Total Map (%)	Unique Map (%)
A-T-1	91.68%	86.58%
A-T-2	92.36%	87.49%
A-T-3	87.76%	82.36%
A-N-1	87.03%	81.72%
A-N-2	89.47%	84.55%
A-N-3	83.18%	78.38%
A-P-1	91.06%	85.92%
A-P-2	87.48%	82.10%
A-P-3	90.83%	85.60%
A-K-1	87.91%	82.61%
A-K-2	89.36%	83.77%
A-K-3	78.54%	66.05%
B-D-1	80.31%	75.17%
B-D-2	84.80%	79.34%
B-D-3	90.88%	86.18%
B-C-1	89.18%	84.01%
B-C-2	88.03%	82.68%
B-C-3	91.49%	86.83%
B-H-1	89.44%	84.46%
B-H-2	91.21%	86.01%
B-H-3	90.37%	85.51%

Note: The unique mapping rate is the proportion of reads uniquely aligned to the genome of *C. hodginsii*, and the mapping rate is the proportion of reads fully aligned to the genome of *C. hodginsii*.

## Data Availability

Data are contained within the article.

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
