# Peer review of "Transcriptomic and Phenotypical Analysis of the Physiological Plasticity of Chamaecyparis hodginsii Roots under Different Nutrient Environments and Adjacent Plant Competition"

_plants, 2024, doi:10.3390/plants13182641_

Round 1

Reviewer 1 Report

Comments and Suggestions for Authors

Title: “Transcriptomic and Phenotypical Analysis of the Physiological Plasticity of Chamaecyparis hodginsii Roots Under Heterogeneous Nutrient Environments and Adjacent Plant Competition”

“Heterogenous”- what does it mean? Deficit or excess stress. It should be understandable from the title and clearly defined in the abstract and other parts of the manuscript too.  

Abstract:

It should be shortened.

The treatments should be included (dose/method).

The results of the later part should be clearly and briefly mentioned.

Introduction

Line 61, 3: et al please give . after al

Correct similarly throughout the manuscript.

Line 94: “Henry et Thomas” not clear.

Line 98: “20 one-year-old” Please write in a different way.

Root architecture, canopy structure, root and shoot growth pattern etc. how can be affected under the growing condition should be discussed.

Line 105-107: “The results lay a theoretical foundation for understanding the molecular regulatory mechanism of the nutrition foraging 106 and adaptive development of C. hodginsii.”

Delete this part. The introduction should be finished with objectives, not with results.

Materials and methods

This section should be decreased in volume.

“Twenty 1-year-old” please write in a similar way-Twenty one-year-old.

Line 126: Twenty 1-year-old seedlings? The age of seedlings, Okay?

Please include the climatic data such as, temperature, precipitation, relative humidity during the growing season.

Please mention unit in the Table 2.

Please cite references in Table  1 and in Table 2.

The materials and methods section is unnecessarily lengthy. The description of the experimental condition is not organized.  

Results:

Figure 2: Mention unit in g-1 FW or g-1 DW (MDA contents).

The unit of antioxidant enzyme activities can be expressed as mg-1 protein. Please check both units.

Discussion

Line 874-875: “invasive species and fungi, which was 874 also similar to the results of this study”. The present study did not focus on pathological data. So, it is better to avoid these statements.

Line 885-886: Without study it better avoid writing “to transmit hormone signals and induce phenylpropanone synthesis”

Line 888-889: “The root activity of C. hodginsii seedlings in N and P heterogeneous nutrient environment was higher than that in K heterogeneous and homogeneous nutrient environment, while it was opposite in K heterogeneous environment.” Why this phenomenon happened/how it is beneficial/harmful-explain/discuss.

Write down/discuss the importance of antioxidant enzyme activities with other data of the present study.

Conclusion

Mention the limitations and future outlook of the study. How the experiment can solve existing problems?

Comments on the Quality of English Language

The manuscript is too lengthy and can be reduced especially the materials and methods section. There are linguistic problems throughout the paper. I suggest language editing by professionals.   

Author Response

Comment1: Abstract should be shortened.The treatments should be included (dose/method).The results of the later part should be clearly and briefly mentioned.

Response1:The abstract has been refined according to the requirements, and the conclusion part has been added.

Comment2: Introduction: Line 61, 3: et al please give . after al; Correct similarly throughout the manuscript; Line 94: “Henry et Thomas” not clear; Line 98: “20 one-year-old” Please write in a different way; Root architecture, canopy structure, root and shoot growth pattern etc. how can be affected under the growing condition should be discussed; Line 105-107: “The results lay a theoretical foundation for understanding the molecular regulatory mechanism of the nutrition foraging 106 and adaptive development of C. hodginsii.”; Delete this part. The introduction should be finished with objectives, not with results.

Response2:It has been modified as required and some contents have been deleted.

Comment3: Materials and methods: This section should be decreased in volume; “Twenty 1-year-old” please write in a similar way-Twenty one-year-old; Line 126: Twenty 1-year-old seedlings? The age of seedlings, Okay?; Please include the climatic data such as, temperature, precipitation, relative humidity during the growing season.; Please mention unit in the Table 2.; Please cite references in Table  1 and in Table 2.; The materials and methods section is unnecessarily lengthy. The description of the experimental condition is not organized. 

Response3:The climate data were supplemented, the expression of seedling age was unified, and the expression was refined.

Comment4: Results: Figure 2: Mention unit in g-1 FW or g-1 DW (MDA contents).;The unit of antioxidant enzyme activities can be expressed as mg-1 protein. Please check both units.

Response4:It has been modified according to the requirements

Comment5: Discussion: Line 874-875: “invasive species and fungi, which was 874 also similar to the results of this study”. The present study did not focus on pathological data. So, it is better to avoid these statements.; Line 885-886: Without study it better avoid writing “to transmit hormone signals and induce phenylpropanone synthesis”; Line 888-889: “The root activity of C. hodginsii seedlings in N and P heterogeneous nutrient environment was higher than that in K heterogeneous and homogeneous nutrient environment, while it was opposite in K heterogeneous environment.” Why this phenomenon happened/how it is beneficial/harmful-explain/discuss.; Write down/discuss the importance of antioxidant enzyme activities with other data of the present study.

Response5:The text expression in the discussion is improved and some contents are added.

Comment6: Conclusion: Mention the limitations and future outlook of the study. How the experiment can solve existing problems?

Response6:The limitations and future prospects of the study have been supplemented.

Reviewer 2 Report

Comments and Suggestions for Authors

The study of Bingjun Li and co-authors entitled ”Transcriptomic and Phenotypical Analysis of the Physiological Plasticity of Chamaecyparis hodginsii Roots Under Heterogeneous Nutrient Environments and Adjacent Plant Competition” is relevant, well-designed and performed using an adequate and modern methods. The obtained results are well-presented and discussed. The authors have done a lot of work and obtained important results. The results provide a theoretical basis for understanding the molecular regulatory mechanisms underlying the nutrition foraging and adaptive development of C. hodginsii.

However, some revision is required:

Abstract  

It is recommended to shorten the text a little, leaving the most important points. The conclusion sentence should be added, describing the importance of obtaining results and possible applications.

Material and Methods

Provide references to the literature for all the methods used (Lines 230-239, 254-257, 262-264, 267-277).

Line 226 – “TTC” write the full name at the first mention.

Results

Figures 3-6, 8-14. The text in figures is difficult to read, it is necessary to improve the quality of presentation.

Conclusion

The conclusion sentence should be added, describing the importance of obtaining results and possible applications.

Other comments:

Abbreviations should be defined the first time they appear in each of three sections: the abstract; the main text; the first figure or table. Check, please.

The additional sections must be added before References: Author Contributions, Funding, Conflicts of Interest, etc. (https://www.mdpi.com/journal/plants/instructions)

Best regards,

Reviewer

Author Response

Comment1: Abstract: It is recommended to shorten the text a little, leaving the most important points. The conclusion sentence should be added, describing the importance of obtaining results and possible applications.

Response1: The abstract text has been shortened and a concluding section has been added

Comment2:Material and Methods: Provide references to the literature for all the methods used (Lines 230-239, 254-257, 262-264, 267-277).;Line 226 – “TTC” write the full name at the first mention.

Response2: The references in the test method section and the full name of TTC have been supplemented.

Comment3:Results;Figures 3-6, 8-14. The text in figures is difficult to read, it is necessary to improve the quality of presentation.

Response3: It has been modified according to the requirements

Comment4:Conclusion:The conclusion sentence should be added, describing the importance of obtaining results and possible applications.

Respomse4:Added conclusions and prospects

Comment5: Abbreviations should be defined the first time they appear in each of three sections: the abstract; the main text; the first figure or table. Check, please.

Respomse5:The content has been added as required
